# MicroRNA Expression Variation in Female Dog (*Canis familiaris*) Reproductive Organs with Age and Presence of Uteropathy

**DOI:** 10.3390/ani12233352

**Published:** 2022-11-29

**Authors:** Eun Pyo Kim, Chae Young Kim, Min Young Heo, Sang Wha Kim, Geon A. Kim

**Affiliations:** 1Department of Theriogenology and Biotechnology, Research Institute for Veterinary Science, College of Veterinary Medicine, Seoul National University, Seoul 08826, Republic of Korea; 2College of Veterinary Medicine, Seoul National University, Seoul 08826, Republic of Korea; 3Department of Biomedical Laboratory Science, School of Healthcare Science, Eulji University, Uijeongbu 34824, Republic of Korea; 4Department of Microbiology and Immunology, Institute of Endemic Disease, Seoul National University College of Medicine, Seoul 08826, Republic of Korea

**Keywords:** *Canis familiaris*, microRNA, uterine disease, aging, reproductive organ

## Abstract

**Simple Summary:**

To analyze miRNA changes for female reproductive aging and organs with uteropathy in dogs, the miRNA expressions of female reproductive organs were compared and validated with real-time PCR.

**Abstract:**

While aging is associated with microRNA (miRNA) expression, little is known about its role in the aging of dog reproductive organs. We examined miRNA expression in ovaries, oviducts, and uteri from young and old dogs and dogs with uteropathy to elucidate miRNA’s role in aging. The ovaries, oviducts, and uteri of 18 dogs (*Canis familiaris*)—young (8.5 ± 1.9 months old), old (78.2 ± 29.0 months old), and those with uteropathy (104.4 ± 15.1 months old)—were collected for miRNA expression examination. Total RNA samples were extracted, reverse-transcribed to cDNA, and real-time PCR analysis was also performed. In ovaries, miR-708 and miR-151 levels were significantly higher in old dogs than in young dogs, and only let-7a, let-7b, let-7c, miR125b, and miR26a were significantly upregulated in dogs with uteropathy. In the oviducts and uteri of old dogs, miR-140, miR-30d, miR-23a, miR-10a, miR-125a, miR-221, and miR-29a were upregulated. Realtime quantitative PCR revealed that targeted mRNA was similarly regulated to miRNA. These results suggest that miRNAs of reproductive organs in dogs may be biological markers for aging and reproductive diseases and could be used for mediating aging.

## 1. Introduction

Aging is a complex process in which environmental and genetic factors can modulate the accumulation of cellular and molecular damage, leading to the functional decline of tissues and organs. Recently, numerous studies have suggested the cellular hallmarks of aging [1]. One of which is epigenetic alterations, which represent a mechanism of aging and age-related disorders. Epigenetic alterations are mediated by three major mechanisms: DNA methylation, histone modification, and regulation by noncoding RNAs. MicroRNAs (miRNAs) are 20–24 nucleotide-long non-coding RNAs that repress translation or induce mRNA degradation of target transcripts with important roles in all biological pathways [2]. Approximately 60% of all human mRNAs are predicted to be controlled by miRNAs [3,4], and the most common mode of post transcriptional gene regulation is the alteration in protein expression due to the miRNA regulation [5]. In mice, the miRNAs of the brain, liver and skeletal muscles have been shown to be differentially expressed with age [6], and in humans, changes in peripheral blood mononuclear cell miRNA expression have also been reported [7,8].

It has been assumed that humans and dogs (*Canis familiaris*) share many attributes, including ageing. Therefore, it has been proposed that domesticated dogs are a powerful model to better understand genetic and environmental determinants of aging [9]. It has become an accepted general rule that female mammals live longer than male mammals. However, dogs did not show age-specific mortality differences between sexes. Among intact animals, there is a pattern for male dogs to live slightly longer than females, but among neutered dogs, females live longer [10]. One of the most common uterine diseases in intact female dogs is pyometra, which affects approximately 50% of the intact population [11]. Pyometra is reported as one of the most common proliferative and inflammatory diseases of the canine endometrium [12], and mainly affects middle-aged or older individuals. However, it has been reported in individuals from 4 months to 18 years, with a mean age of 6–8 years [11]. In Sweden, an average of 20% of all bitches are diagnosed with pyometra before 10 years of age [13]. Although the complex pathogenesis of pyometra including hydrometra and mucometra is not completely understood, it involves both hormonal imbalances and bacterial infections. Resistance to bacterial infections in the reproductive tract often depends on the innate immune system of the uterus [14]. A study on miRNA expression in LPS-induced endometritis bovine endometrial cells suggested that the inflammatory cascade affects miRNAs [15]. Prior studies have also indicated correlation between miRNAs and inflammatory diseases such as pneumonia [16], mastitis [17] and endometritis [18].

While there have been previous studies on miRNAs in dogs with diseases such as cardiac hypertrophy [19], and cervical spondylomyelopathy [20], there has not been a study on miRNAs in female reproductive organs from individuals of varying ages and disease statuses. In this study, we aimed to compare the expression profiles of miRNAs in the ovaries, oviducts, and uteri from dogs of different ages to dogs with uteropathy. We hypothesized that (1) the miRNAs of reproductive organs would be differentially expressed according to age and disease status and (2) that transcript expression complementary to the miRNAs would provide similar results.

## 2. Materials and Methods

### 2.1. Collection of Tissue Samples

Ovaries, oviducts, and uteri were collected with the owners’ permission from 18 female dogs at local animal hospitals that needed general neutering operations or ovariohysterectomies (OHE) due to diagnosed pyometra. Before the OHE procedure, all animals underwent accurate clinical assessments, such as blood chemistry tests and complete blood cell counts, to detect any possible pathological statuses and examine whether they were suitable for general anesthesia. In the case of uterine disease, all cases included in this study had vaginal discharge, dilation into the uterus was confirmed on ultrasonography, and uterine tissue was cut longitudinally to the lumen to confirm the uterine cavity’s pus. Immediately after surgery, the ovaries, oviducts, and uterus of each dog were isolated. The ovarian bursa and its ligaments were removed in order to collect the ovaries and oviducts. The bifurcation region of the uterus was used for analysis without uterine blood vessels or the broad ligament. Only ovarian, oviduct, and uterine tissues of each dog were used for RNA extraction and stored at −80 °C for further gene expression analysis.

### 2.2. RNA Isolation and Quality Check of RNA and cDNA Synthesis

Total RNA was extracted from samples of each reproductive organ (at least 50 mg), and was homogenized with 1 mL of RNA lysis solution using the easy-spin™ Total RNA Extraction Kit (Intron Biotechnology, Seoul, Republic of Korea) per the manufacturer’s instructions. For gene microarray hybridization, RNA integrity and quantity were evaluated using a 2100 Bioanalyzer (G2939BA, Agilent, CA, USA) with an Agilent RNA 6000 Nano Kit. Only samples conforming to the conditions (A260/A280 and A260/A280; >1.0, concentration; >50 ng/µL, volume; >10 µL, total amount; >0.7 ug, rRNA ratio; >1.0, RIN; >7.0 with visible small RNA peaks) were subjected to microRNA analysis. The quality and quantity of the RNA were measured using a NanoDrop 2000 spectrophotometer (Thermo Fisher Scientific Inc., Wilmington, MA, USA), and then the RNA samples (500 ng) were reverse-transcribed to cDNA (total volume of 20 µL) using the Maxime RT-PCR premix kit (Intron Biotechnology, Seoul, Republic of Korea).

### 2.3. Gene MicoRNA Hybridization, Scanning, and Data Processing

All protocols were conducted as described in the standard Affymetrix Expression Analysis Technical Manual (Affymetrix, Inc., Santa Clara, CA, USA). RNA was labeled using the FlashTag™ Biotin HSR RNA Labeling Kit (Thermo Fisher, Waltham, MA, USA). Biotin-labeled samples were hybridized to the GeneChip^®^ Affymetrix miRNA microarray (Affymetrix). All arrays were scanned using the Affymetrix GeneChip^®^ scanner, and raw data analysis was performed using the Affymetrix GeneChip^®^ Command Console^®^ Software (AGCC). The raw data images generated from the scanner were processed into CEL files containing the measured intensities for each probe on the array. A fold change of >2 and *p*-value of <0.05 was used as a cut-off to screen a wide range of differentially expressed miRNAs.

### 2.4. Real-Time PCR Quantification

Real-time PCR analysis was performed using a StepOnePlus Real-Time PCR System (Applied Biosystems, Waltham, MA, USA, #4376600) with 2 × SYBR Green PCR Master Mix (Applied Biosystems, Waltham, MA, USA, #4309155). Relative gene expression results were normalized to the reference gene β-actin (ACTB), which has been recommended as a stable and common housekeeping gene in canine reproductive tissues (Du et al., 2016), then expressed in arbitrary units. The reactions were performed according to the manufacturer’s protocol. To evaluate the effect of miRNA regulation, we first selected miRNAs and searched for target genes using the MicroRNA Target Prediction Database, TargetScan (https://www.targetscan.org/cgi-bin/targetscan/vert_71/, accessed on 1 July 2021). The gene names, accession numbers of the 14 selected genes, and their primer sequences (Bioneer, Daejeon, Republic of Korea) are listed in Table 1. The reaction parameters were as follows: initial enzyme activation step for 10 min at 95 °C; 40 cycles of 10 s at 95 °C, 20 s at 60 °C, and 40 s at 72 °C. Cooling was carried out for 30 s at 40 °C. Each sample was amplified three times. Relative quantification of gene expression levels was assessed using the ΔΔCt method: relative quantity (R)=2−(ΔCt sample − ΔCt control).

### 2.5. Statistical Analysis

The clinical data of the dogs in this study are presented as mean ± standard deviation. Comparative analysis among the groups was performed using the unpaired *t*-test, and statistical analysis was performed using the SPSS software (version 25.0; SPSS Inc., Chicago, IL, USA). Statistical significance was set at the less < 0.05.

## 3. Results

The clinical trial results of the female dogs used in this study are shown (Table 2). The breeds of healthy young females under 1 year of age (average 8.5 ± 1.9 months) were: mixed (1), shiba inu (1), pomeranian (1), maltese (1), and poodle (2). The average hematological data was as follows: total protein (6.2 ± 0.3 g/dL), glucose (116.3 ± 5.9 mg/dL), blood urea nitrogen (BUN, 19.9 ± 1.2 mg/dL), creatine (0.6 ± 0.1 mg/dL), white blood cell (WBC, 10.3 ± 0.8 (10^9^/L)), red blood cell (RBC, 7.0 ± 0.2 (10^12^/L)), hematocrit (HCT, 47.6 ± 1.0%), platelet (277.3 ± 36.7 (10^9^/L)), alkaline phosphatase (ALKP, 198.3 ± 24.0 U/L), and alanine aminotransferase (ALT, 52.2 ± 5.8 U/L).

The breeds of healthy females above the age of 1 year (average 78.2 ± 29.0 months) were: poodles (1), bichon frise (1), shih tzu (2), chihuahua (1), and maltese (1). The hematological data was: total protein (6.1 ± 0.2 g/dL), glucose (109.3 ± 2.3 mg/dL), BUN (20.5 ± 2.9 mg/dL), Creatine (0.6 ± 0.1 mg/dL), WBC (10.6 ± 0.5 (10^9^/L)), RBC (7.1 ± 0.2 (10^12^/L)), HCT (48.8 ± 1.7%), Platelet (348.5 ± 36.8 (10^9^/L)), ALKP (176.0 ± 9.3 U/L), ALT (62.3 ± 6.7 U/L).

There were six breeds of females with pyometra (104.4 ± 15.1 months): mixed (3), shih tzu (1), spitz (1), and yorkshire terrier (1). The mean hematological data were: total protein (7.1 ± 0.4 g/dL), glucose (115.0 ± 14.1 mg/dL), BUN (24.1 ± 14.8 mg/dL), creatine (0.7 ± 0.3 mg/dL), WBC (12.1 ± 4.2 (10^9^/L)), RBC (7.2 ± 0.8 (10^12^/L)), HCT (48.3 ± 6.7%), platelet (324.7 ± 195.8 (10^9^/L)), ALKP (359.5 ± 211.8 U/L), ALT (46.8 ± 48.9 U/L). The reference range of each hematological level is as follows. total protein: 5~7.2 (g/dL); glucose: 75~128 (mg/dL); BUN: 9.2~29.2 (mg/dL); creatine: 0.4~1.4 (mg/dL); WBC: 5~20 (10^9^/L); RBC: 5.5~8.5 (10^12^/L); HCT (Hematocrit): 35~56 (%); platelet: 100~500 (10^9^/L); ALKP: 47~254 (U/L); and ALT: 17~78 (U/L). The hematological results of healthy bitches both under and over 1 year of age did not deviate from the reference ranges. However, bitches with pyometra presented with high levels of BUN, platelets, ALKP, and ALT, though they were still considered for general anesthesia during ovariohysterectomy surgery at a local animal hospital. Patients diagnosed with pyometra were those whose pus was discharged outside of the vagina and only those who were clearly diagnosed by the ultrasonography and then checking the pus in the uterine cavity during ovariohysterectomy were included in this study.

The miRNA volume plots for comparison between immature dogs younger than 1 year and mature dogs older than 3 years of age are illustrated in Figure 1. With a fold change cut-off of >1.5 (up or down regulation) and a *p*-value of <0.05, two miRNAs (miR-151 and miR-708) were found to be significantly upregulated in the ovarian tissue of mature dogs compared to that of immature dogs. In the oviducts, miR-30d and miR-140 were highly expressed in mature dogs. In the uterus, the top 5 significant miRNAs were identified by considering the volume value, fold change, and *p*-value. All five miRNAs (miR-29a, miR-125a, miR-23a, miR-10a, and miR-221) were overexpressed in the uterus of elderly dogs compared to in young dogs.

The volume plots of miRNA for comparison between mature dogs above 3 years of age and older dogs with uteropathy are shown in Figure 2. Using the same aforementioned criteria, 49 significant miRNAs (1 downregulated, 48 upregulated miRNAs— navy dots) were identified in the ovaries. The expression of let-7a, let-7b, let7c, miR-125b, and miR-26a in the ovaries of mature dogs with uteropathy was relatively higher than in those of healthy dogs over 3 years of age. However, the expression of miR-203 was lower in the oviducts of dogs with uteropathy than in those of healthy dogs over 3 years of age. No significant differences were found in the miRNA expression in the uteri.

Finally, to identify potential target molecules that might be directly affected by the upregulation of miRNAs, we determined in silico common targets of the upregulated miRNAs. Fourteen genes were identified, 14 gene primer pairs for real-time PCR gene quantification analysis were designed, then the gene expression levels were compared (Figure 3). The relative expression levels of the predicted target APH1A of miR151 and MPRS35 of miR708 was significantly lower in mature ovaries than in immature ovaries. In the oviducts, there were no significant differences between the predicted target genes of miR30d and miR140. In uteri, miR-23a predicted the target AUH, whose expression in mature uteri was lower than in immature uteri. To determine whether miRNA expression changes were correlated with their targeted mRNA dysregulation, three targeted genes were selected. Although ARPP19 expression as predicted by miR-26 was higher in uteropathy afflicted ovaries, STRADB predicted by miR26a, and E1F1AD predicted by miR125b presented with no significant differences between the groups.

## 4. Discussion

Although it is widely recognized that aging is a key factor in explaining infertility, to the best of our knowledge, no data on the miRNA expression profiles of reproductive organs of dogs of various ages have been published. Therefore, we aimed to explore the miRNA profiles of reproductive organs including the ovaries, oviducts, and uteri from immature, mature, and uteropathy afflicted dogs. The age-related differences in miRNA expression in each tissue were analyzed by dividing the dogs into two age groups: younger than 1 year, and older than 3 years. In addition, to examine the effect of uteropathy on miRNA expression, the same reproductive organs of individuals with and without uteropathy were compared. The diagnostic criterion for uteropathy was whether there was an increase in echo in the sonographs of dogs who came to the local veterinary hospital with vaginal discharge.

Cellular senescence from aging presents as various epigenetic changes, for which there are three main mechanisms: DNA methylation, histone modifications, and regulatory microRNAs (miRNAs) [21]. miRNAs are classified according to their size— some are small non-coding RNAs involved in gene regulation as post-transcriptional regulators, and others are elements of chromatin-modifying complexes. Our study confirmed that the expression of various types of miRNAs in female reproductive organs are affected by age and uterine diseases.

The miR-151 predicted target, APH1A (anterior pharynx-defective 1), is a component of γ-secretase. γ-secretase is not only able to cleave type-I transmembrane proteins, but also the amyloid precursor protein and Notch receptors [22]. Alterations in the Notch pathway are significantly associated with poor clinical outcomes in patients with ovarian cancer [23], and therefore we can speculate that the expression of miR-151, APH1A and the notch3 pathway may be correlated with the ovaries and associated with aging. One human study examined miRNAs in serum from young and old individuals. The small RNA sequencing and real-time RT-PCR results showed increased levels of miR-151 in old individuals [8]. In our study, the results of miR-151 also showed a significant increase, as expected from the target gene APH1A transcript expression.

The microRNA-30 (miR-30) family contains five members and six mature miRNA molecules (miR-30a, miR-30b, miR-30c-1, miR30c-2, miR-30d, and miR-30e), and affects the expression of recombinant proteins by regulating the ubiquitin E3 ligase-Skp2-induced ubiquitin pathway, thereby affecting the development of the reproductive system [24]. Interestingly, one study using canine oviductal cells and their microvesicles found that miR-30b with miR-375 and miR-503, which target factors such as WNT, MAPK, neurotrophin, and ubiquitin, could be detected and were involved in follicular growth and oocyte maturation [25]. Similarly, our study found that miR-30d was among the significant miRNAs that were more highly expressed in the oviducts of older dogs.

It has been shown that the levels of miR-7, miR-468, miR-542, and miR-698 increased in mouse muscles due to skeletal muscle aging, whereas miR-124a, miR-181a, miR-221, miR-382, miR-434, and miR-455 decreased [26]. Interestingly, miR-221 was more highly expressed in the uterus, which is the representative muscle tissue of the female organ in dogs older than 3 years of age. It has is known that miR-221 along with miR-222 in mice can promote vascular smooth muscle cell calcification [27], and we thus assumed that miR-221 regulates smooth muscle aging. Furthermore, aged uteri in dogs showed upregulated miR-125a in fold changes and volcano plots in our study. It has been shown that miR-125, a well-conserved homolog of lin-4, and co-transcribed in Drosophila, has been shown to modulate lifespan in a sex- and tissue-specific manner [28]. Moreover, our miR-125 results from uteri of dogs appears consistent with those of a study showing that miR-125 overexpression in neurons reduces female Drosophila lifespan [28].

When miRNA expression in younger (31 ± 2 years old) and older (73 ± 3 years old) human skeletal muscle was compared, 18 miRNAs were found to be differentially expressed, with let-7b and let-7e being upregulated in the older muscles [29]. However, let-7a, let-7b, and let-7c in the ovaries of dogs with uteropathy were upregulated compared to in healthy dog ovaries. Interestingly, in an in vitro inflammatory model using bovine endometrial epithelial cells, let-7c was found to play an important role for regulating inflammation [15]. One study found that overexpression of this miRNA reduced uterine inflammation [30] and another study showed that let-7c reduced the release of proinflammatory cytokines [31]. Unlike our findings, the results of these previous studies suggest that the let-7 family serves as an anti-inflammatory mimic. However, if the function of let-7 is assumed to be also anti-inflammatory in the ovaries of dogs, it can be assumed that the ovarian let-7 family, including let-7a, let-7b, and let-7c, increases as a compensatory response to reduce or repair uterine inflammation or pathological status. According to prior literature, the role of ovaries in controlling inflammatory diseases in the uterus is unclear. However, our study demonstrated that the let-7 expression in the ovaries of dogs with an inflammatory uterine disease was significantly higher than in healthy ovaries, presumably to provide a specific signal for the inflammation of the uterus. These results indicate that further studies on the effects of let-7 on the uterus are needed.

## 5. Conclusions

This study had several limitations. Firstly, the breeds of healthy dogs across the age groups did not match. Secondly, the estrous cycle of each dog was not evaluated. In human studies, miRNA expression has been shown to fluctuate, such as in the levels of the two main classes of endocrine hormones from the follicular, ovulation, and to the luteal phase [32]. Thirdly, only dogs with vaginal discharge for uteropathy were included in this study. On histological examination, cystic endometrial hyperplasia—defined as proliferation of endometrial glands—endometrial hyperplasia, and cyst formation, can present with or without fluid accumulation in the uterus. Uterine diseases can also be classified into mucometra, hydrometra, and pyometra according to the type of fluid present in the uterus and the degree of mucin. Although pus in the uterus found from the OHE indicated pyometra and thus which individual’s tissue to use, the etiological pathogen could be either *Escherichia coli*, *Staphylococcus spp*., or *Streptococcus* [13]. The exact pathogens and associated miRNAs need to be clarified in further studies. The last limitation is the small sample size. Large-scale follow-up studies are required to examine the effects of age and the most common female reproductive disease, pyometra, on miRNA expression.

This study investigated miRNA expression variation with age and uteropathy. With further research based on our study, the identified miRNAs will provide clues and therapeutic targets for delaying aging, improving fertility, and preventing uterine diseases.

## Figures and Tables

**Figure 1 animals-12-03352-f001:**
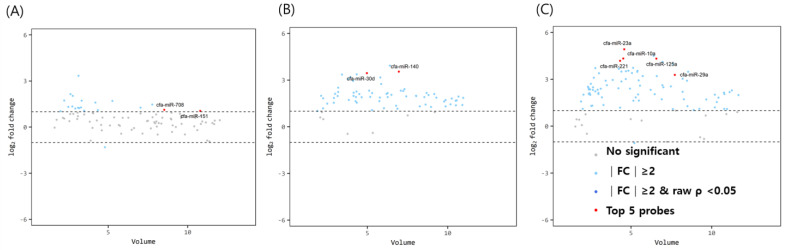
Volume plots of miRNAs in (**A**) ovary, (**B**) oviduct and (**C**) uterus of above 3 years dogs compared with below 1 years old dogs. The volcano plot shows the fold change and according volume of the miRNAs between the dogs with healthy young and old dogs. The horizontal line displayed a 2-fold change differences. Gray dots mean no significance for comparison between volume and fold change value. Sky blue dots mean above 2-fold change differences without *p* < 0.05. cfa-miR = *Canis lupus familiaris* microRNA. Red dots means significant top five miRNAs for comparisons.

**Figure 2 animals-12-03352-f002:**
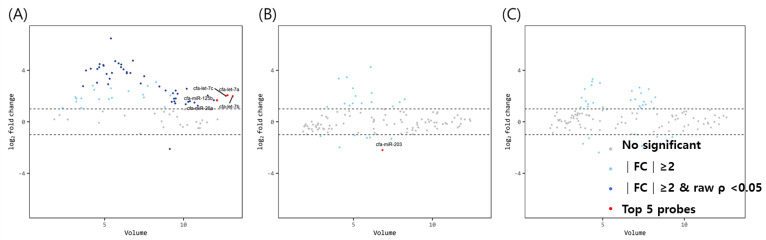
Volume plots of miRNAs in (**A**) ovary, (**B**) oviduct and (**C**) uterus of dogs older than 3 years old with uteropathy compared to healthy dogs older than 3 years old. The volcano plot shows the fold change and according volume of the miRNAs between the two groups. The horizontal line displayed a 2-fold change differences. Gray dots mean no significance for comparison between volume and fold change value. Sky blue dots mean above 2-fold change differences without *p* < 0.05. Navy blue represents significant fold change and *p* < 0.05. cfa-miR = Canis lupus familiaris microRNA. Red dots mean significant top five miRNAs for comparisons.

**Figure 3 animals-12-03352-f003:**
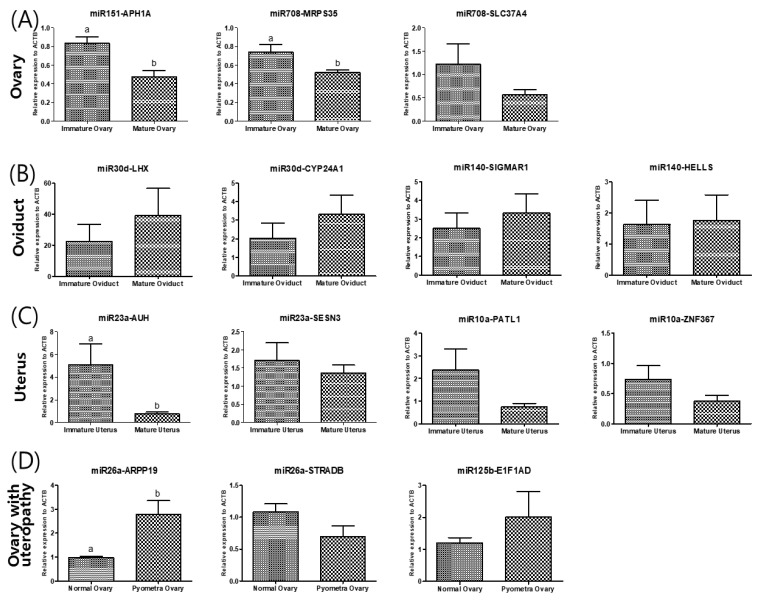
Validation of the differentially expressed levels of each predicted of miRNA of (**A**) ovary, (**B**) oviduct, (**C**) uterus and (**D**) ovary of dogs with uteropathy. (A) The differentially expressed miR in ovary was cfa-miR-151 and cfa-miR-708. The cfa-miR-151 is expected to target APH1A, cfa-miR-708 target MRPS35 and SLC37A4. (B) The differentially expressed miR of oviduct is cfa-miR-30d and cfa-miR-140. Their predicted target is LHX8 and CYP24A1 of cfa-miR-30d and SIGMAR1 and HELLS of cfa-miR-140. (**C**) The differentially expressed miR in uterus was cfa-miR-23a and cfa-miR-10a. The cfa-miR-23a is expected to target SESN3 and AUH, cfa-miR-10a target PATL1 and ZNF367. (**D**) The differentially expressed miR of ovary with uteropathy is cfa-miR-26a and cfa-miR-125b. Their predicted target is ARPP19 and STRADB of cfa-miR-26a and E1F1AD of cfa-miR-125b.The gene expression was validated by performing quantitative real-time PCR. The data is shown as mean ± SD, *p* < 0.05. cfa-miR = *Canis lupus familiaris* microRNA.

**Table 1 animals-12-03352-t001:** Predicted target of miRNA and primer sequence of each genes.

MicroRNA	Ortholog of Target Gene	Gene Name	Ori	Primer Sequence	Accession No.
cfa-miR-151	APH1A	APH1A gamma secretase subunit	F	cctactacaagctgcttaag	XM_038423042.1
			R	gataacagagaagacaccac	
cfa-miR-708	MRPS35	mitochondrial ribosomal protein S35	F	gcacgagtagtaaccttaag	NM_001284487.1
			R	ctgtctgttgtgatggtaag	
	SLC37A4	solute carrier family 37 (glucose-6-phosphate transporter), member 4	F	gttgtctccttcctctgt	NM_001287131.2
			R	gtaatgtactctcgtccttc	
cfa-miR-30d	LHX8	LIM homeobox 8	F	cacatccattctactgactg	XM_038670394.1
			R	tctgcagaggactttctc	
	CYP24A1	cytochrome P450, family 24, subfamily A, polypeptide 1	F	gtatactgctggcttacttg	XM_038434121.1
			R	ggacaggtacatttagtgac	
cfa-miR-140	SIGMAR1	sigma naon-opioid intracellular receptor 1	F	cagactcacataccacaag	XM_038681024.1
			R	ccagacagagtataataccc	
	HELLS	helicase, lymphoid-specific	F	gagaaagaagagaggaagag	XM_038439975.1
			R	cacagagattagaagaggag	
cfa-miR-23a	SESN3	sestrin 3	F	ctgtgtttccctactgtatc	XM_038429728.1
			R	ctgttgactgagaggaatac	
	AUH	AU RNA binding methylglutaconyl-CoA hydratase	F	gatatacgtgtagcagcttc	XM_038655406.1
			R	gtgcagagaagatgagct	
cfa-miR-10a	PATL1	PAT1 homolog 1, processing body mRNA decay factor	F	cttcctaccttctgtgttac	XM_038424237.1
			R	ctacggtctactctgaactg	
	ZNF367	zinc finger protein 367	F	cataggctgctattctgtag	XM_038654936.1
			R	ctgtactagaagtcccgtat	
cfa-miR-26a	ARPP19	cAMP-regulated phosphoprotein, 19kDa	F	ttggatccaggcatcttttc	XM_038442363.1
			R	gtgggtggagcaggaagata	
	STRADB	STE20-related kinase adaptor beta	F	ggacatgcacaggactcaga	XM_038447612.1
			R	tcgggaattcttcattctgg	
cfa-miR-125b	EIF1AD	eukaryotic translation initiation factor 1A domain containing	F	ggctgagatctcctttgtgc	XM_038424930.1
			R	tcctgatgactgtggctctg	

**Table 2 animals-12-03352-t002:** Clinical examination of dogs included in this study.

**Variables** **(Mean ±** **SEM)**	**Healthy Dogs** **below 1 year** **(n = 6)**	**Healthy Dogs** **above 3 year** **(n = 6)**	**Uteropathy** **(n = 6)**	**Reference Value**
Age, months	8.5 ± 1.9	78.2 ± 29.0	104.4 ± 15.1	
Breed	mixed, shiba inu, pomeranian, maltese, poodle	poodle, bichon frise, shih tzu, chihuahua, maltese	mixed, shih tzu, spitz, yorkshire terrier	
Total protein	6.2 ± 0.3	6.1 ± 0.2	7.1 ± 0.4	5~7.2 (g/dL)
Glucose	116.3 ± 5.9	109.3 ± 2.3	115.0 ± 14.1	75~128 (mg/dL)
BUN	19.9 ± 1.2	20.5 ± 2.9	24.1 ± 14.8	9.2~29.2 (mg/dL)
Creatine	0.6 ± 0.1	0.6 ± 0.1	0.7 ± 0.3	0.4~1.4 (mg/dL)
WBC	10.3 ± 0.8	10.6 ± 0.5	12.1 ± 4.2	5~20 (10^9^/L)
RBC	7.0 ± 0.2	7.1 ± 0.2	7.2 ± 0.8	5.5~8.5 (10^12^/L)
HCT	47.6 ± 1.0	48.8 ± 1.7	48.3 ± 6.7	35~56 (%)
Platelet	277.3 ± 36.7	348.5 ± 36.8	324.7 ± 195.8	100~500 (10^9^/L)
ALKP	198.3 ± 24.0	176.0 ± 9.3	359.5 ± 211.8	47~254 (U/L)
ALT	52.2 ± 5.8	62.3 ± 6.7	46.8 ± 48.9	17~78 (U/L)

BUN, blood urea nitrogen; WBC, white blood cell; RBC, red blood cell; HCT, hematocrit; ALKP, alkaline phosphatase; ALT, alanine aminotransferase.

## Data Availability

Not applicable.

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
