# Peer review of "MicroRNA Expression Variation in Female Dog (Canis familiaris) Reproductive Organs with Age and Presence of Uteropathy"

_animals, 2022, doi:10.3390/ani12233352_

Round 1

Reviewer 1 Report

The paper presents comparisons concerning microRNA expression in young and old bitches and bitches suffering from pyometra.

The paper is well organized and well written, but its methodology raises serious concerns.

The presented qualification of experimental dogs with pyometra is not within the clinical rules commonly approved for such dogs and this disease. Bitch with pyometra is commonly presented with some level of systemic inflammatory response, which reflects the clinical findings such as: changes in demeanour, elevated rectal temperature, mucus membrane colour, as well as biochemical changes (raised inflammatory parameters and sometimes also kidney and liver enzymes), and finally diagnosis confirmation is usually done with the use of US imaging. None of these was included in the presented methodology. The US findings showed only a slightly enlarged uterus, but without the typical pyometra appearance, as with the blood test results, only slightly elevated liver enzymes were reported.

All of the above in my opinion limits the chances of a justified qualification of experimental dogs with pyometra in this study and hence did not allow a fair interpretation of the results and discussion.

Author Response

Question : The presented qualification of experimental dogs with pyometra is not within the clinical rules commonly approved for such dogs and this disease. Bitch with pyometra is commonly presented with some level of systemic inflammatory response, which reflects the clinical findings such as: changes in demeanour, elevated rectal temperature, mucus membrane colour, as well as biochemical changes (raised inflammatory parameters and sometimes also kidney and liver enzymes), and finally diagnosis confirmation is usually done with the use of US imaging. None of these was included in the presented methodology. The US findings showed only a slightly enlarged uterus, but without the typical pyometra appearance, as with the blood test results, only slightly elevated liver enzymes were reported.

Answer : First of all, thank you for your detailed and good point. We also thought a lot about screening cases for pyometra. It would be great if all pyometra patients had the same symptoms and hematological changes, the degree of disease (such as the nature of the exudate and the degree of uterine dilatation), but, all patients do not have the same clinical symptoms, or always have the same changes.  

Patients with pyometra used in this study could be screened as samples only when clear vaginal discharge was identified or the uterus filled with suppurative exudation was clearly identified with ultrasound definite uterine expansion. Samples must be able to confirm that the uterus removed after surgical operation was clearly pyometra with clear pus in the uterus. Also the individuals had at least two other symptoms such as vomiting, loss of appetite, high fever, hematological abnormalities, etc).

Meanwhile, as reviewer_1 pointed out, the picture in Figure 1. B used in the paper is the uterus of an open type pyometra of a small dog, so we would like to remove Figure 1 and add the comments “. Patients diagnosed with pyometra were those whose pus was discharged outside of the vagina and only those who were clearly diagnosed by the ultrasonography and then checking the pus in the uterine cavity during ovariohysterectomy were included in this study.” In line 136~140.

Thank you.

Question : All of the above in my opinion limits the chances of a justified qualification of experimental dogs with pyometra in this study and hence did not allow a fair interpretation of the results and discussion.

Answer : As the reviewer said, this study has the disadvantage that all conditions of all dogs (breed, age, diet, environment) are not the same in that it is not an experimental dog, but companion dogs living with humans.

Reviewer 2 Report

This is a well written, informative paper. I suspect that the results will contribute to the understanding of senescence of female reproductive organs and the pathophysiology of pyometra, which will open up new avenues for treatment down the road. I like that the authors appropriately address the limitations of the study. One of the most difficult things to reconcile in such studies is where to draw the line between age groups. For example, in this study are listed poodle, Shiba Inu, and mixed breed that are  less than a year. However, a toy or miniature poodle will be mature at 11 months, whereas a standard poodle is not. Some of the less than a year old dogs will have gone into heat already, whereas other have not. Maybe the solution would be to sort the dogs either by age at puberty or closure of growth plates, but I realize that is not possible at this stage. These are  just some  general observations.

I have only a few minor comments. 

Line 73: Was the Affymetrix chip in this study canine specific?

Lines 90 and half of 91: It looks like there is a font/size difference in the lettering.

Figure 1: Not really needed but nice illustration. If you keep this figure, I would recommend outline the normal uterus in (A), as for the untrained eye, it looks like uterine loops instead of an empty uterus. 

Line 179: It should primer pairs not just primers.

Line 220: "...sequencing results..." Did you mean sequencing? Aren't these based off of chip analysis? 

Line 221: "...the result of miR-151 also showed a significant result..." This  should be reworded to reflect the increase (i.e. ...showed a significant increase). 

Line 246: It is not clear if the up regulation occurs in the younger or older population. 

I would replace vaginal exudate with vaginal discharge throughout the document, as this is the more common terminology.

Author Response

Question : This is a well written, informative paper. I suspect that the results will contribute to the understanding of senescence of female reproductive organs and the pathophysiology of pyometra, which will open up new avenues for treatment down the road. I like that the authors appropriately address the limitations of the study. One of the most difficult things to reconcile in such studies is where to draw the line between age groups. For example, in this study are listed poodle, Shiba Inu, and mixed breed that are  less than a year. However, a toy or miniature poodle will be mature at 11 months, whereas a standard poodle is not. Some of the less than a year old dogs will have gone into heat already, whereas other have not. Maybe the solution would be to sort the dogs either by age at puberty or closure of growth plates, but I realize that is not possible at this stage. These are just some general observations.

Answer: Thank you for your positive feedback and good comments. In fact, all of the individuals used in this paper are small dogs. Small dogs don't all show the same development, growth, or physiology, but they don't generally make much difference between them like large dogs like standard poodles. That's why we used only small dogs as samples. In this study, in the case of dogs under 1-year-old, ovariohysterectomy was performed before the first estrus.

I have only a few minor comments. 

Question : Line 73: Was the Affymetrix chip in this study canine specific?

Answer: Yes, we used the Affymetrix chip for canine specific.

Question : Lines 90 and half of 91: It looks like there is a font/size difference in the lettering.

Answer: I will modify the font/size of the part that you pointed out in line 90-91.

Question : Figure 1: Not really needed but nice illustration. If you keep this figure, I would recommend outline the normal uterus in (A), as for the untrained eye, it looks like uterine loops instead of an empty uterus. 

Answer: As you pointed out, figure 1 is not really needed. So we would like to remove.

Question : Line 179: It should primer pairs not just primers.

Answer:  I will modify "primers" to "primer pairs" in line 176.

Question : Line 220: "...sequencing results..." Did you mean sequencing? Aren't these based off of chip analysis?

Answer:  This study use small RNA sequencing and real-time RT-PCR for miRNA analysis in line 216-217

Question : Line 221: "...the result of miR-151 also showed a significant result..." This  should be reworded to reflect the increase (i.e. ...showed a significant increase). 

Answer: I’ll change from "significant result" to "significant increase" in line 218.

Question : Line 246: It is not clear if the up regulation occurs in the younger or older population. 

Answer: The reference mentions that all of the let7 series of miRNAs have increased in the elderly individual, and I will change the English sentence to clearly indicate that this part is also increasing in old age in line 243.

Question : I would replace vaginal exudate with vaginal discharge throughout the document, as this is the more common terminology.

Answer: Overall, I will replace the vaginal exudate with the vaginal discharge in the paper in line 202 and 264. I understood your advice and correction. I will revise all the comments you gave me and submit them. Thank you very much for the detailed correction.

Reviewer 3 Report

Autor (y) sugerujÄ…, że miRNA narzÄ…dów rozrodczych u psów mogÄ… być biologicznymi markerami starzenia siÄ™ i chorób reprodukcyjnych i mogÄ… być stosowane do poÅ›redniczenia w starzeniu siÄ™.

Ovaries, oviducts, and uteri were collected with the owners’ permission from 18 female dogs at local animal hospitals that needed general neutering operations or 46 ovariohysterectomies (OHE) due to diagnosed pyometr, but was there approval from the animal ethics committee for such a study? Were the younger female dogs after their first heat? Was the number of heat exchanged analyzed in the remaining bitches? With such a small group, however, cannot different breeds of dogs have an impact on the results, especially since there is a difference in the amount and course of the heat itself in each breed and possible complications after them?

In my opinion, the resulting differences may have an impact on the results obtained in the study investigated miRNA expression variation with age and pyometra and therapeutic targets for delaying aging, improving fertility, and preventing inflammatory diseases!

Author Response

Question : Ovaries, oviducts, and uteri were collected with the owners’ permission from 18 female dogs at local animal hospitals that needed general neutering operations or 46 ovariohysterectomies (OHE) due to diagnosed pyometra, but was there approval from the animal ethics committee for such a study? 

Answer : This study secured and analyzed female genital tissue for early diagnosis and prevention of reproductive diseases in dogs. In particular, this study was conducted on dogs of owners who voluntarily visited animal hospitals, and accordingly, there was no choice but to secure limited varieties and ages of animals under study. The pet was approved by dog owners who visited the animal hospital at a local animal hospital outside Eulji University and used samples of the reproductive organs of ordinary households, not individuals in the laboratory.

In general, when visiting an animal hospital for pyometra or spay surgery, the tissue is discarded and no invasive or additional necessary treatment is required for this study.

In addition, samples were secured by clinical experts at individual veterinary hospitals outside Eulji University and Seoul National University.

We received a confirmation letter from the head of Eulji University's Animal Experimental Ethics Committee regarding the exemption from the experimental animal approval and submitted it to Animals.

Question : Were the younger female dogs after their first heat? Was the number of heat exchanged analyzed in the remaining bitches?

Answer : Female dogs under a year old did not have their first heat, and the rest of them did not count the number of heat exchange.

Question : With such a small group, however, cannot different breeds of dogs have an impact on the results, especially since there is a difference in the amount and course of the heat itself in each breed and possible complications after them?

Answer: The individuals used in this study selected small dog individuals of similar age groups and sizes. All (except for those with pyometra) identified and recruited healthy and non-specific problems in preoperative tests, and those with pyometra also confirmed that there were no other diseases or only minor changes except for pyometra.

However, as you pointed out, I think the size of the group and the variety of individuals and other health conditions could have affected the results of the study. I think that part needs to be supplemented with further research after this paper.

Question : In my opinion, the resulting differences may have an impact on the results obtained in the study investigated miRNA expression variation with age and pyometra and therapeutic targets for delaying aging, improving fertility, and preventing inflammatory diseases!

Answer : Thank you for agreeing that miRNA expression is an important key to aging, disease, and treatment as the gist of our paper.

Round 2

Reviewer 1 Report

The explanations provided by the authors still do not convince me. Pyometra is a condition with typical clinical symptoms, regardless of the patient. Systemic symptoms must be present in addition to the purulent content of the uterus to establish that it is a pyometra case.The patients presented in the study had uterine dilation of unknown origin and, in my opinion, cannot be classified as pyometra cases.

Furthermore, changing the description of the patent qualification will not be suitable in this case. This immediately raises another question: how the authors recognized the pus, was it the purulent discharge and not just thick mucus in the investigated cases?

The authors might consider classifying these bitches as females suffering from some degree of uteropathy or uterine fluid accumulation without general symptoms. Because these were the clinical findings seen during classification of patients, in the presented study. At the same time, classifying them as pyometra cases is, in my opinion, an overinterpretation.

If the authors consider changing the principles of patient eligibility, the hypothesis and discussion require reassessment and re-interpretation to make them consistent with the studied groups of patients.

Author Response

Although the veterinarians in the animal hospitals has clearly diagnosed all collected samples, at this point it is not possible to distinguish and check all uterine disease case. So, as reviewer recommended, we would like to replace the pyometra with uteropathy. In addition, since M&M lacked detailed explanations on how to diagnose these uterine diseases including mucometra, hydrometra and pyometra, we added the following explanation in line 51-54. Thanks to the reviewer.

: In the case of uterine disease, all cases included in this study had vaginal discharge, dilation into the uterus was confirmed on ultrasonography, and uterine tissue was cut longitudinally to the lumen to confirm the uterine cavity’s pus.    

Due to your comments, our manuscript has been improved by changing the term to a more precise expression. 
